# Exploring the Role of Intraoperative Positive Culture of Allograft Bone in Subsequent Postoperative Infections among Donors and Recipients in Bone Bank Processing

**DOI:** 10.3390/diagnostics14010015

**Published:** 2023-12-20

**Authors:** Yao-Hung Tsai, Hung-Yen Chen, Tsung-Yu Huang, Jiun-Liang Chen, Liang-Tseng Kuo, Kuo-Chin Huang

**Affiliations:** 1Department of Orthopaedic Surgery, Chia-Yi Chang Gung Memorial Hospital, Chiayi 61363, Taiwan; s0059508@cgmh.org.tw (H.-Y.C.); yq01393@cgmh.org.tw (J.-L.C.); 8902078@adm.cgmh.org.tw (L.-T.K.); kc2672@adm.cgmh.org.tw (K.-C.H.); 2College of Medicine, Chang Gung University at Taoyuan, Taoyuan 33302, Taiwan; r12045@cgmh.org.tw; 3Division of Infectious Diseases, Department of Internal Medicine, Chia-Yi Chang Gung Memorial Hospital, Chiayi 61363, Taiwan

**Keywords:** allograft, bone bank, retrieval, discard rate, donor, recipient

## Abstract

Background: Allografts have been frequently used in orthopedic procedures. The purposes of this study were to evaluate the discard rates and bacterial contamination of a bone bank, and to assess the clinical outcomes of recipients with bacterial culture-positive donor allografts. Methods: We retrospectively reviewed 1764 allografts which were harvested from living donors and stored in a bone bank from 2018 to 2022. The donors whose allografts displayed bacterial contamination at retrieval of the primary hip or knee arthroplasty were followed for microbiology and subsequent prosthetic joint infection analysis. The infected pathogens, antibiotic treatment and subsequent infection were reviewed for the intraoperative positive culture group. Results: The discard rate was 17%, and the bacterial contamination rate of bone retrieval was 2.15%. Thirty-eight allografts at retrieval displayed confirmed bacterial growth, and 37 patients did not reveal infective signs at 6 months follow-up. A total of 1464 allografts were stored and implanted, among which 28 allografts (1.91%) were confirmed to be positive for bacterial growth and 13 cases (0.89%) were confirmed as surgical site infections. Conclusions: Our results validate the suggestion that our bone bank system performs good quality monitoring to eliminate the risk of dissemination of viral and bacterial diseases and to decrease surgical site infection after allograft implantation. By ensuring aseptic conditions and contamination-reducing strategies during harvesting and thawing, the allografts can be safely stored and implanted while limiting bacterial contamination. Our findings confirm that the intraoperative positive cultures of allografts did not contribute to subsequent postoperative surgical site infection in donors and recipients.

## 1. Introduction

Bone allografts are frequently used in surgery, providing a structural framework and osteoconduction in many orthopedic procedures, including revision knee and hip arthroplasty, spine surgery, acute fracture with bone loss, and nonunions [1,2,3]. Bone allografts can be obtained from living or deceased donors and then stored in a bone bank; however, the operative procedures from retrieval to transplantation must be performed under aseptic conditions and a hygienic environment to prevent bacterial contamination [1,3,4,5].

Bone banks are institutions responsible for the procurement and storage of bones from living or cadaveric donors. The sterile procurement of bone allografts and the strict screening of donors are mandatorily performed in bone banks based on the criteria provided by the Centers for Disease Control and Prevention (CDC), which can reduce the risk of disease transmission and bacterial contamination in the recipients [6,7,8,9]. The bacterial contamination of the implanted allografts can result in a high incidence of prosthetic joint infections, infected nonunions, and wound infection [10,11]. The contamination rates of allografts at retrieval have been reported as being up to 22%, and the rates of positive intraoperative bone culture in recipients receiving allograft transplantation have been reported as being high as 12% [1,2,8,10,11,12]. Nevertheless, some studies have reported that the positive intraoperative cultures of retrieval and implanted allografts are not significantly associated with postoperative surgical site infection [1,9,10,11,12,13].

The purposes of this study were to evaluate the discard rates and the bacterial contamination of bone bank processing, to investigate the surgical site infection rate in patients with bacterial growth of allografts at the time of retrieval and implantation, and to assess the clinical outcomes of recipients with bacterial culture-positive donor allografts following bone transplantation.

## 2. Materials and Methods

### 2.1. Study Design

We retrospectively reviewed 1764 allografts from the hospital bone bank registration system, including 806 femoral heads from total hip arthroplasty (THA) or bipolar hemiarthroplasty and 958 resected bone chips from total knee arthroplasty (TKA), which were harvested from living donors and stored in the bone bank from January 2018 to December 2022 in Chia-Yi Chang Gung Memorial Hospital. Routine serological blood tests checking for hepatitis B (HBs-antigen), hepatitis C (anti-HCV antibody), human immunodeficiency virus (HIV 1/2 antibodies) and syphilis (Venereal disease reference laboratory test, VDRL) were performed. The wound cultures were obtained with sterile culturette swabs during surgery, and the microbiologic results were confirmed a few days after surgery. Those allografts which revealed any positive results for bacterial growth, hepatitis B, hepatitis C, VDRL or HIV were discarded. Patients who underwent allograft implantation surgery were included in this study. The positive intraoperative bacterial cultures of harvesting and thawing allograft were also analyzed.

### 2.2. Allograft Retrieval and Bone Banking Process

The bone bank of Chia-Yi Chang Gung Memorial Hospital was established in 2009, authorized and evaluated every three years by the Taiwan Food and Drug Administration (TFDA) of the Ministry of Health and Welfare. The quality assessment of the bone bank, such as the retrieval process, the cleanliness of the environment and the quality assurance of the operating room, the storage of allografts in a refrigerator at −70 °C to −80 °C, and the implantation process, is performed under the practice guidelines of the Taiwan FDA.

Allograft retrieval from living donors was performed under the sterile conditions of an operating room with informed consent. The donors received preoperative prophylactic antibiotics within one hour of the skin incision, and were given at least one day of postoperative antibiotics. The retrieved allografts were swabbed for bacteriological cultures from the bone surface and then soaked in cefazolin solution at a concentration of 2 mg/dL for at least 10 min after checking the bone quality. After the removal of soft tissues and cartilage, the allograft bones were put into two certified antifreezing plastic bags and wrapped in a sterile drape. The bagged allograft was wrapped with another drape with a label attached, and was placed in a bigger plastic bag to avoid the label becoming detached (Figure 1). The wrapped allograft was stored in the unproved layer of refrigerator at first, and was moved to the confirmed layer by the bone bank superintendent after the negative results of bacterial growth and four serological blood tests were confirmed. The uncertificated allografts were moved to another freezer and discarded.

### 2.3. Allograft Implantation

After checking the information of the donor, the allograft was removed from the freezer for thawing for 20 min before it was needed for implantation. The wrapped allograft was opened and swabbed for bacterial cultures by the scrub nurse. Then, the allograft bones were soaked in warm sterile cefazolin solution with a minimum bactericidal concentration of 2 mg/dL until it was implanted. All recipients received preoperative prophylactic antibiotics and were administered postoperative antibiotics for at least one day. If the cultures revealed positive bacterial growth, extended intravenous or oral antibiotics according to the antimicrobial sensitivity of pathogens were given.

### 2.4. Microbiology Laboratory Procedures

Isolates of pathogens from primary culture were identified by colonial appearance, Gram stain, agglutination with specific antisera, and conventional biochemical tests used in clinical microbiology laboratories. Matrix-assisted laser desorption/ionization-time of flight mass spectrometry (MALDI-TOF MS) was used to identify these isolates to the species level.

The antimicrobial susceptibility of pathogens was performed by the hospital microbiology laboratory via the standard disk diffusion technique. These antimicrobial susceptibility tests were performed as recommended by the Clinical and Laboratory Standards Institute (CLSI), and the results were interpreted according to the CLSI criteria for these microorganisms.

### 2.5. Clinical Assessment

The donors whose allografts revealed bacterial contamination at the retrieval of primary hip or knee arthroplasty were followed for microbiology and subsequent prosthetic joint infection analysis. The medical records and postoperative wound conditions of the included donors and recipients were reviewed for at least 6 months. According to the culture results of the implanted allografts, the patients were divided into the intraoperative positive culture group and the intraoperative negative culture group. Demographic data, such as gender, age, allograft types, underlying chronic diseases, operative procedures for allograft implantation, infected pathogens and antibiotic treatment, were reviewed for the intraoperative positive culture group. To assess those clinical outcomes, surgical site infection was defined as wound complications or surgical management required within 6 months after implantation.

### 2.6. Statistical Analysis

Statistical analyses were performed with the use of Statistical Product and Service Solutions (SPSS) Version 18.0 statistical software (SPSS, Chicago, IL, USA). We used the Fisher exact test for categorical variables to examin intraoperative positive culture of allograft related to postoperative surgical site infections between the intraoperative positive culture group and the intraoperative negative culture group. A value of *p* < 0.05 was considered to indicate statistical significance. 

## 3. Results

A total of 1764 allografts were retrieved in a five-year period, and 300 allografts were excluded because they revealed positive results of hepatitis B, hepatitis C, VDRL, HIV or bacterial growth. The total discard rate was 17% (300/1764), and the bacterial contamination rate of bone retrieval was 2.15% (38/1764). A total of 1464 allografts were included in this study. In total, 644 femoral head allografts (44%) and 820 TKA allografts (56%) were implanted, and intravenous and oral antibiotics were administered after surgery (Figure 2).

The most common causes for discarding allografts were positive serological tests of living donors, which revealed anti-HCV antibodies in 136 sample, HB antigens in 106 samples, both hepatitis B and C in 5 cases, HIV in 3 cases, and VDRL in 12 cases. Meanwhile, 38 allografts at bone retrieval, including 22 femoral head allografts and 16 TKA bone chip allografts, displayed confirmed bacterial growth, among which *Staphylococcus epidermidis* (23.7%) was the major pathogen (Table 1). Thirty-seven patients did not reveal infective signs at 6 months follow-up, and one patient with *Klebsiella pneumonia* and *Micrococcus* infection of the retrieved allograft developed a prosthetic joint infection with *Viridans streptococcus* in the 10th month after TKA (1/38, 2.63%).

A total of 1464 allografts were stored in our bone bank and underwent implantation, and 13 cases of surgical site infections were confirmed (0.89%). A total of 28 allografts (1.91%), 11 of femoral heads and 17 of TKA bone chips, were confirmed as being positive for bacterial growth in wound cultures a few days later and were included in the intraoperative positive culture group. The most common contaminant pathogen was *Staphyococcus epidermidis* (8/28, 28.6%), followed by *Staphylococcus haemolyticus* (4/28, 14.3%). One patient (case 25) was found to have postoperative superficial wound breakage in the 4th month, and underwent local debridement which revealed no bacterial growth in the wound culture (Table 2).

A total of 1436 patients displayed a negative intraoperative culture of the implanted allografts, and 12 patients had surgical site infections. Seven patients had a previous prosthetic joint infection and had recurrent infection after performing revision surgery and allograft implantation. Four patients who underwent fracture fixation surgery with allograft implantation received debridement due to wound infections. One patient with spine lumbar fusion surgery was managed with a superficial wound infection. There were no significant differences regarding postoperative surgical site infections between the intraoperative positive culture group and the intraoperative negative culture group (*p* = 0.223) (Table 3).

## 4. Discussion

Allografts are frequently used in orthopedic reconstructive procedures, and successful outcomes after allograft implantation depend on the quality control of bone bank processing. The bone bank can store allografts from living or cadaveric donors and safely supply them for recipients, and must prevent the transmission of infectious diseases and malignancies through routine culture swabs and serological screening [2,4,5,7]. However, there are no uniform guidelines for the management of bone banks, and most bone bank protocols for quality control follow the guidelines recommended by the American Association of Tissue Banks and also meet the national law standards [1,3,6,14,15,16]. The discard rates reported in the literature vary from 5% to 46% [8,14,17,18]. Baseri et al. systematically reviewed 12 studies and found that the bacterial contamination rate at the time of retrieval of allografts from living donors was 7.5% [17]. In this study, the total discard rate of our bone bank within a 5-year interval was 17%, and the bacterial contamination rate of bone retrieval from living donors was 2.15%, which could effectively eliminate the risk of dissemination of viral and bacterial diseases. Finally, 1464 allografts were stored and implantation was performed, and 13 cases (0.89%) of surgical site infections were confirmed, which means that our bone bank implemented good quality monitoring under the supervision and certification of the Taiwan FDA.

The bacterial contamination rates of bone allografts at the time of retrieval vary from 0.13% to 28.5%, and femoral head allografts from living donors are commonly reported to have a contamination rate from 0% to as high as 22% [3,4,10,17,19]. Coagulase-negative *Staphylcoccus* (CoNS) was the most commonly isolated contaminant pathogen [3,4,10,17,19,20]. However, many studies have demonstrated that living donors with an intraoperative culture-positive femoral head allograft were not significantly associated with prosthetic joint infection and long-term failure of THA [20,21,22,23]. Justesen et al. reported that 12% of TKA patients displayed an intraoperative culture-positive finding, and none developed clinical infection within the first year after primary TKA [24]. Jonsson et al. had presented intraoperative contamination was common in the THA and TKA, but few prosthetic joint infections occurred [25]. Although 97.3% (37/38) patients with bacterial contamination at allograft retrieval did not develop wound complications or prosthetic joint infection in this study, we still need to pay attention to following those donors with intraoperative bacterial growth of the discarded allografts and consider extending the postoperative antibiotic administration according to the donors’ clinical conditions.

The overall positive bacterial culture rate of after-thawing allografts before implantation ranged from 1% to 12%, and the infection rate in recipients with contaminated allograft was reported as being from 0% to 10% [1,2,5,6,11,12,13]. However, the infective pathogens of postoperative surgical site infections were different from the cultured microorganisms of thawed allografts [1,2,5,6,11,12,13]. Sims et al. reported that 43 implanted allografts (43/996, 4.3%) had positive bacterial growth, and significant postoperative infection developed in two patients (4.6%) [1]. Barahart et al. reported that eight recipients (8/230, 3.5%) were found to have a positive intraoperative bacterial culture of the implanted allograft, and one (12.5%) developed an infection [13]. Both authors stated that routine cultures of implanted allografts are not necessary because subsequent postoperative infections are rare, and thus several centers in Canada no longer perform intraoperative allograft culture for cost-saving purposes [1,13]. Although we found only one recipient in the intraoperative positive culture group with subsequent superficial wound infection (3.6%), and no significant differences regarding postoperative surgical site infections between the intraoperative positive culture group and the intraoperative negative culture group, we still support applying the swabbed bacterial culture method to thawed allografts to supervise their sterility in bone transplantation and maintain the quality control of bone bank processing.

The microorganisms cultured during allograft harvesting and implantation were more likely to have resulted from skin flora, airborne bacteria, the environment, surgical wounds, instruments, and contaminated gloves of the surgeon or scrub nurse in the operation room [12,20,26,27]. The methodologies of sterilization for the allograft, such as irradiation, heat, ethylene oxide and soaking with antimicrobial solutions, were considered to decrease the chances of allograft contamination [2,4,9]. However, soaking the allograft in antimicrobial solutions at retrieval and transplantation is commonly performed and has been proven to be effective in the operating room due to the lower exposure time and low cost in the literature [2,5,9]. Therefore, we chose it as a standard sterilization method.

Deijekers et al. described that the microorganisms found on the contaminated allografts were split into groups with low pathogenicity (such as CoNS, *Corynebacterium* spp. and *Propionibacterium* spp.) and high pathogenicity (such as *Streptococcus*, *Staphylococcus aureus*, *Escherichia coli*, and *Clostridium*), and they indicated that the main contaminating microorganisms were those bacteria with low pathogenicity [9]. CoNS species were the most commonly reported pathogens appearing at the retrieval and thawing of the allografts; however, those microorganisms in the intraoperative positive cultures did not correlate with those pathogens in the subsequent surgical site infection [1,2,5,9,10,11,12,13,14,26]. Our study also demonstrated that CoNS was found in 60.5% (23/38) of allografts at retrieval and 53.6% (15/28) of allografts at thawing, and observed that the pathogen in the subsequent infection of both patients (one donor and one recipient) was not the same as the bacteria in the culture-positive allografts. Therefore, the improvement of contamination-reducing strategies is a priority for guaranteeing the efficiency of quality system monitoring in bone bank processing. Routine prophylactic antibiotic use preoperatively and postoperatively, rinsing the allograft with an antibiotic solution, and changing gloves before handling the allograft were recommended to reduce the pathogenic microbial load and the relative risk of subsequent infection [2,5,9,11,13,14,25,26]. By following these strategies, the subsequent postoperative infection rates with contaminated allografts in donors and recipients were found to be low in our study (2.6% and 3.6%, respectively).

One strength of this study is that we prospectively collected the medical records of both donors and recipients in our bone bank registration system, and traced the clinical condition of those donors and recipients whose allograft had bacterial contamination at 6 months follow-up after a surgical site infection. We compared the subsequent postoperative infection rates between the intraoperative positive culture group and the intraoperative negative culture group in recipients with allograft implantation, which was seldom mentioned and analyzed in the literature. Another strength of our work was presenting the kind and duration of postoperative antibiotic use in positive bacterial culture recipients, which may provide relevant information for surgeons regarding whether to extend the prophylactic antibiotic administration.

This study has several limitations. First, we did not include cadaveric donors. Although our institution is a tertiary care hospital which is located on the western coast of southern Taiwan, we were unable to obtain cadaveric donors smoothly because most of the residents are old-aged and have multiple underlying chronic diseases. Second, we performed the four serological screening tests after surgery. Most of the living donors did not know whether they had hepatitis B, hepatitis C, syphilis or HIV infection before performing the joint replacement. We may decrease the discarding rate by prescreening the four serological tests.

## 5. Conclusions

Our results validate the suggestion that our bone bank system conducts good quality monitoring to eliminate the risk of dissemination of viral and bacterial diseases and to decrease surgical site infections after allograft implantation through strict screening protocols and routinely swabbing cultures. By ensuring aseptic conditions and implementing contamination-reducing strategies during harvesting and thawing, the allografts can be safely stored and implanted while limiting bacterial contamination. Our findings confirm that the intraoperative positive cultures of allografts did not contribute to subsequent postoperative surgical site infection in the donors and recipients.

## Figures and Tables

**Figure 1 diagnostics-14-00015-f001:**
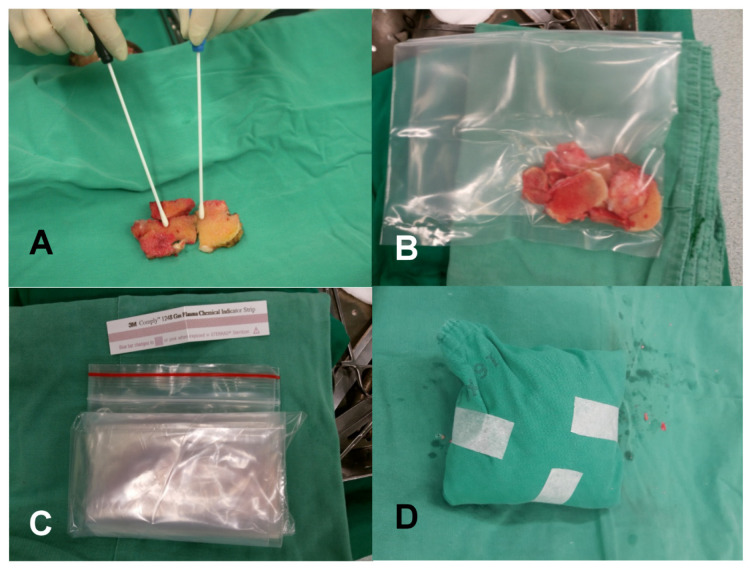
The bagging and wrapping process after allograft retrieval. (**A**) The allografts were swabbed for bacteriological cultures. (**B**) The allograft bones were put into first certified antifreezing plastic bag. (**C**) This bag was put into the second bigger bag. (**D**) The plastic bag was wrapped in a sterile drape.

**Figure 2 diagnostics-14-00015-f002:**
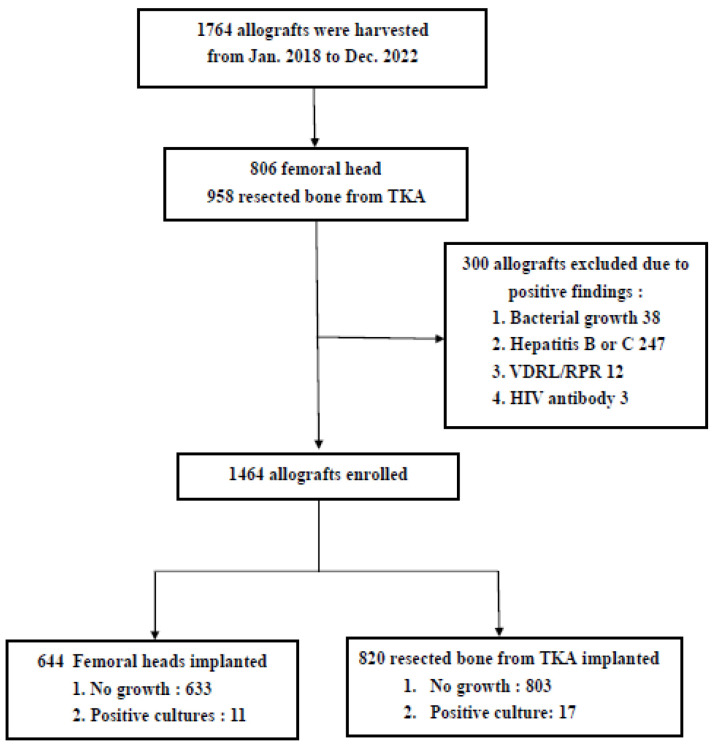
Flow chart of allografts’ inclusion.

**Table 1 diagnostics-14-00015-t001:** Microbiology of contaminated allografts at retrieval in donors.

Variable	Number of Allografts	Prosthetic Joint Infection
*Staphylococcus epidermidis*	9	0
*Staphylococcus haemolyticus*	4	0
*Staphylococcus capitis*	3	0
*Staphylococcus caprae*	2	0
*Staphylococcus hominis*	1	0
CoNS	4	0
MSSA	3	0
MRSA	2	0
*Klebsiella pneumonia* and *Micrococcus*	1	1
*Clostrium perfringens*	1	0
*Peptococcus*	1	0
*Peptoniphilus harei*	1	0
*Propionibacterium acnes*	1	0
*Enterococcus faecium*	1	0
*Bacillus cereus*	2	0
Gram-positive bacilli	2	0
	38	1

CoNS, coagulase-negative staphylcoccus; MSSA, methicillin-sensitive *Staphylococcus aureus*; MRSA, methicillin-resistant *Staphylococcus aureus*.

**Table 2 diagnostics-14-00015-t002:** Characteristics and outcomes of the recipients with intraoperative culture-positive allografts.

Number	Sex	Age	Allograft Type	Underlying Chronic Disease	Operative Procedure	Microorganism	Intravenous Antibiotics	IVA Duration	Oral Antibiotics Duration	Results	Preoperative Condtion
1	M	78	Femoral head	Heart disease, Gout	Revision THA	*Staphylococcus haemolyticus*	Cefazolin	7	7	N	N
2	M	81	Femoral head	HT	Lumbar spine surgery	*Staphylococcus haemolyticus*	Cefazolin	14	0	N	N
3	M	58	Femoral head	Alcoholism	Revision THA	Gram-positive bacilli	Cefazolin	1	0	N	PJI-*Staph. epidermidis*
4	F	67	Femoral head	Heart disease	TKA	Gram-negative bacilli	Cefazolin	1	0	N	N
5	F	70	Femoral head	HT	Humeral fracture	*Staphylococcus epidermidis*	Cefazolin	7	14	N	N
6	M	62	Femoral head	HT	High tibial osteotomy	*Propionibacterium* spp.	Cefazolin	1	0	N	N
7	M	55	Femoral head	DM, HT, ESRD, CAD	Ankle fusion	*Enterococcus faecium*, MM	Cefazolin	7	14	N	N
8	F	64	Femoral head	DM, HT, ESRD	Femoral fracture	*Staphylococcus epidermidis*	Cefazolin	3	0	N	N
9	M	77	Femoral head	HT, HB, COPD	Thoracic spine surgery	*Staphylococcus epidermidis*	Cefazolin	7	14	N	N
10	M	55	Femoral head	DM, ESRD	Revision THA	MRSA	Teicoplanin	7	60	N	PJI-MRSA
11	F	71	Femoral head	DM, HT	Revision TKA	*Staphylococcus saprophytica*	Cefazolin	7	0	N	N
12	F	57	TKA bone chips	DM, HT	Tibial fracture	*Staphylococcus epidermidis*	Cefazolin	2	0	N	N
13	M	78	TKA bone chips	Heart disease, Gout	Revision THA	*Staphylococcus haemolyticus*	Cefazolin	7	7	N	N
14	F	63	TKA bone chips	DM, HT, Gout	Lumbar spine surgery	*Stenotrophomonas maltophilia*	Cefazolin	14	0	N	N
15	F	78	TKA bone chips	DM, HT	Lumbar spine surgery	*Enterococcus faecium*	Cefazolin	7	14	N	N
16	F	59	TKA bone chips	DM, HT	Lumbar spine surgery	Gram-positive bacilli	Cefazolin	3	0	N	N
17	F	67	TKA bone chips	DM, HT	Patellar fracture	*Staphylococcus haemolyticus*	Cefazolin	1	0	N	N
18	M	58	TKA bone chips	HT	THA	*Staphylococcus epidermidis*	Ceftazidime	14	0	N	Pneumonia- *Pseudomonas*
19	F	76	TKA bone chips	HT	Radial fracture	*Staphylococcus epidermidis*	Cefazolin	1	0	N	N
20	M	66	TKA bone chips	HB, Heart disease	Lumbar spine surgery	*Staphylococcus capitis*	Cefazolin	3	0	N	N
21	F	65	TKA bone chips	HT	Calcaneus fracture	*Staphylococcus epidermidis*	Cefazolin	7	0	N	N
22	M	63	TKA bone chips	HCC, LC	Revision THA	CoNS	Cefazolin	1	14	N	N
23	M	68	TKA bone chips	HB, HC	Thoracic spine surgery	*Staphylococcus epidermidis*	Cefazolin	7	7	N	N
24	F	50	TKA bone chips	HB	Tibial fx	*Bacillus cereus*	Cefuroxime	14	0	N	N
25	M	68	TKA bone chips	DM, Cancer	Humeral fracture	*Enterococcus faecium*	Cefazolin	2	7	Y	N
26	F	81	TKA bone chips	HT	Radial fracture	*Aerococcus viridans*	Cefazolin	1	0	N	N
27	M	56	TKA bone chips	HT, ESRD, HC	Lumbar spine surgery	*Bacillus flexus*	Flomocef	7	0	N	N
28	F	61	TKA bone chips	HB	Lumbar spine surgery	*Micrococcus luteus*	Flomocef	7	7	N	N

HCC, hepatic cell carcinoma; LC, liver cirrhosis; HB, hepatitis B; HC, hepatitis C; DM, diabetes mellitus; HT, hypertension; CAD, coronary artery disease; ESRD, end stage renal disease; COPD, chronic obstructive pulmonary disease. MRSA, methicillin-resistant *Staphylococcus aureus*; MM, *Morganella morganii*; CoNS, coagulase-negative staphylococcus. TKA, total knee arthroplasty; THA, total hip arthroplasty; IVA, intravenous antibiotics; PJI, prosthetic joint infection.

**Table 3 diagnostics-14-00015-t003:** Comparison between the intraoperative positive culture group and the intraoperative negative culture group for postoperative surgical site infections.

	Intraoperative Positive Culture Group (N = 28)	Intraoperative Negative Culture Group (N = 1436)	*p* Value
No infections	27	1424	0.223
Infection	1	12	

## Data Availability

The datasets used and/or analyzed during the current study are available from the author (orma2244@adm.cgmh.org.tw) upon reasonable request.

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
