# Peer review of "Exploring the Role of Intraoperative Positive Culture of Allograft Bone in Subsequent Postoperative Infections among Donors and Recipients in Bone Bank Processing"

_diagnostics, 2023, doi:10.3390/diagnostics14010015_

Round 1

Reviewer 1 Report

Comments and Suggestions for Authors

Not all conclusions are equally valid. You write

1. "Our results validate that our bone bank system operates a good quality monitoring to eliminate the risk of dissemination of viral and bacterial disease by strict screening protocols and routine swabbed cultures" Please explain what data support this conclusion because reading the manuscript it seems to be only an opinion.

2. "The bacterial contamination can be limited by ensuring aseptic conditions and contamination-reducing strategies during allografts harvesting and thawing" Once more what results support this conclusion? Again it seems to be only an opinion. 

Please explain it or reconstruct conclusions,

Comments on the Quality of English Language

Moderate editing of English language required.

Author Response

Answers for reviewer 1:

I really appreciate your review and positive comment on my manuscript.

  1. "Our results validate that our bone bank system operates a good quality monitoring to eliminate the risk of dissemination of viral and bacterial disease by strict screening protocols and routine swabbed cultures" Please explain what data support this conclusion because reading the manuscript it seems to be only an opinion.
  • Answer: The data of 13 cases (0.89%) with surgical site infections may explain the conclusion.

(1) I have added a sentence on page 5, second paragraph: A total of 1464 allografts were stored in our bone bank and were performed implantation, and 13 cases were confirmed surgical site infections (0.89%).

(2) I have rewritten the sentence in the discussion section on page 9:  In this study, the total discard rate of our bone bank within a 5-year interval was 17% and the bacterial contamination rate of bone retrieval from the living donors was 2.15%, which could effectively eliminate the risk of dissemination of viral and bacterial diseases. Finally, 1464 allografts were stored and performed implantation, and 13 cases (0.89%) were confirmed surgical site infections, which means that our bone bank had a good quality monitoring under the supervision and certification of Taiwan FDA.

(3) I have rewritten the conclusion on Page 1 & 11: Our results validate that our bone bank system operates a good quality monitoring to eliminate the risk of dissemination of viral and bacterial diseases and to decrease the surgical site infection after allograft implanation by strict screening protocols and routine swabbed cultures.

  1. "The bacterial contamination can be limited by ensuring aseptic conditions and contamination-reducing strategies during allografts harvesting and thawing" Once more what results support this conclusion? Again it seems to be only an opinion. 

Please explain it or reconstruct conclusions,

  • Answer: I have rewritten the results and conclusion section in Abstrast on page 1: A total of 1464 allografts were stored and implanted, in which 28 allografts (1.91%) were confirmed positive bacterial growth and 13 cases (0.89%) were confirmed surgical site infections. Conclusions: Our results validate that our bone bank system operates a good quality monitoring to eliminate the risk of dissemination of viral and bacterial diseases and to decrease the surgical site infection after allograft implanation. By ensuring aseptic conditions and contamination-reducing strategies during harvesting and thawing, the allografts can be safely stored and implanted with limiting the bacterial contamination.

Thank you for your kind help

Best regards,

Yao Hung Tsai, M.D.

Dec 16, 2023

Reviewer 2 Report

Comments and Suggestions for Authors

I commend the authors for their exemplary research, titled "Exploring the Role of Intraoperative Positive Culture of Allograft Bone in Subsequent Postoperative Infection among Donors and Recipients in Bone Bank Processing." This retrospective study aims to assess discard rates and bacterial contamination in bone bank processing, investigate the surgical site infection rate in patients with bacterial growth of allografts during retrieval and implantation, and evaluate the clinical outcomes of recipients with bacterial culture-positive donor allografts following bone transplantation. The study's subject matter is compelling, and the manuscript is both clear and readable. The introduction is well-crafted, the results are presented clearly, the discussion is robust, and the conclusions are well-supported by the results. The inclusion of photographs illustrating typical graft preparation effectively enhances the overall comprehensibility of the manuscript.

However, certain crucial aspects of this manuscript require further clarification before it can be deemed suitable for publication. In the materials and methods section, the authors state, "The retrieved allografts were swabbed for cultures and then soaked in cefazolin solution at a concentration of 2 mg/dL at least 10 minutes after checking the bone quality." Some bone banks opt to send the first soaking solution (without antibiotics) for cultures. This approach allows for the easier detection of contamination in allografts compared to using swabs alone. Please provide additional details and rationale for the chosen methodology to swab allografts before soaking in cefazolin solution.

In summary, this manuscript holds promise; however, addressing the aforementioned points will undoubtedly enhance its contribution to the field and increase its suitability for publication.

Author Response

Answers for reviewer 2:

  1. I commend the authors for their exemplary research, titled "Exploring the Role of Intraoperative Positive Culture of Allograft Bone in Subsequent Postoperative Infection among Donors and Recipients in Bone Bank Processing." This retrospective study aims to assess discard rates and bacterial contamination in bone bank processing, investigate the surgical site infection rate in patients with bacterial growth of allografts during retrieval and implantation, and evaluate the clinical outcomes of recipients with bacterial culture-positive donor allografts following bone transplantation. The study's subject matter is compelling, and the manuscript is both clear and readable. The introduction is well-crafted, the results are presented clearly, the discussion is robust, and the conclusions are well-supported by the results. The inclusion of photographs illustrating typical graft preparation effectively enhances the overall comprehensibility of the manuscript.
  • Answer: I really appreciate your review and positive comment on my manuscript. I have changed the title to "Exploring the Role of Intraoperative Positive Culture of Allograft Bone in Subsequent Postoperative Infection among Donors and Recipients in Bone Bank Processing."

  1. However, certain crucial aspects of this manuscript require further clarification before it can be deemed suitable for publication. In the materials and methods section, the authors state, "The retrieved allografts were swabbed for cultures and then soaked in cefazolin solution at a concentration of 2 mg/dL at least 10 minutes after checking the bone quality." Some bone banks opt to send the first soaking solution (without antibiotics) for cultures. This approach allows for the easier detection of contamination in allografts compared to using swabs alone. Please provide additional details and rationale for the chosen methodology to swab allografts before soaking in cefazolin solution.
  • Answer: I have added a sentence on page 9 and marked with yellow lines: The possibilities of microorganisms cultured during allograft harvesting and implantation were more likely to have resulted from skin flora, airborne bacteria, environment, surgical wounds, instruments, and contaminated gloves of the surgeon or scrub nurse in the operation room [12,20,26,27]. The methodologiies of sterilization for the allograft, such as irradiation, heat, ethylene oxide and soaking with antimicrobial solutions, were considered to decrease the chances of allograft contamination [2,4,9]. However, soaking the allograft in the antimicrobial solutions at retrieval and transplantation was commonly performed and was proved effectively in the operating room due to less exposure time and low cost in the literatures [2,5,9]. Therefore, we choose it as a standard sterilization method.

  1. In summary, this manuscript holds promise; however, addressing the aforementioned points will undoubtedly enhance its contribution to the field and increase its suitability for publication.
  • Answer: I am appreciated that the reviewer 2 agrees with our view point.

Thank you for your kind help

Best regards,

Yao Hung Tsai, M.D.

Dec 16, 2023